# Negative Association of Lignan and Phytosterol Intake with Stress Perception during the COVID-19 Pandemic—A Polish Study on Young Adults

**DOI:** 10.3390/nu16030445

**Published:** 2024-02-02

**Authors:** Agnieszka Micek, Paweł Jagielski, Izabela Bolesławska, Anna Maria Witkowska, Anna Waśkiewicz, Zbigniew Wajda, Anna Kamińska, Aneta Cebula, Justyna Godos

**Affiliations:** 1Statistical Laboratory, Jagiellonian University Medical College, 31-126 Kraków, Poland; 2Department of Nutrition and Drug Research, Institute of Public Health, Faculty of Health Sciences, Jagiellonian University Medical College, 31-066 Kraków, Poland; paweljan.jagielski@uj.edu.pl; 3Department of Bromatology, Poznan University of Medical Sciences, 60-806 Poznań, Poland; ibolesla@ump.edu.pl; 4Department of Food Biotechnology, Medical University of Bialystok, 15-295 Bialystok, Poland; anna.witkowska@umb.edu.pl; 5Department of Epidemiology, Cardiovascular Disease Prevention and Health Promotion, National Institute of Cardiology, 04-628 Warszawa, Poland; mwaskiewicz@poczta.onet.pl; 6Faculty of Management and Social Communication, Institute of Applied Psychology, Jagiellonian University, 30-348 Kraków, Poland; zbigniew.wajda@uj.edu.pl; 7Doctoral School of Medical and Health Sciences, Jagiellonian University Medical College, 31-121 Kraków, Poland; anna.kaminska@doctoral.uj.edu.pl (A.K.); aneta.cebula@doctoral.uj.edu.pl (A.C.); 8Department of Biomedical and Biotechnological Sciences, University of Catania, 95123 Catania, Italy; justyna.godos@unict.it

**Keywords:** β-sitosterol, lignan, nutrition, perceived stress, phytosterol, polyphenol, PSS

## Abstract

Background: There has been an increasing global prevalence of depression and other psychiatric diseases in recent years. Perceived stress has been proven to be associated with psychiatric and somatic symptoms. Some animal and human studies have suggested that consuming foods abundant in lignans and phytosterols may be associated with lower levels of stress, depression, and anxiety. Still, the evidence is not yet strong enough to draw firm conclusions. Thus, we investigated the association between dietary intake of these phytochemicals and the level of stress experienced by adult individuals. Methods: Diet was assessed using self-reported 7-day dietary records. The intakes of lignans and phytosterols were estimated using databases with their content in various food products. The Perceived Stress Scale (PSS) was implemented to measure the level of perceived stress. A logistic regression analysis was used to test for associations. Results: The odds of elevated PSS were negatively associated with dietary intake of total phytosterols, stigmasterol, and β-sitosterol, with evidence of a decreasing trend across tertiles of phytochemicals. The analysis for doubling the intake reinforced the aforementioned relationships and found protective effects against PSS for total lignans, pinoresinol, and campesterol. Conclusions: Habitual inclusion of lignans and phytosterols in the diet may play a role in psychological health. To address the global outbreak of depression and other mental health issues triggered by stress, it is important to take a holistic approach. There is a need to develop effective strategies for prevention and treatment, among which certain dietary interventions such as consumption of products abundant in lignans and phytosterols may play a substantial role.

## 1. Introduction

Over the last decades, awareness of mental health issues has been growing, and many organizations and governments have been working to provide better support and resources for those experiencing mental stress [1]. However, this disorder is a social phenomenon that affects an increasing number of people of all ages globally, also as a consequence of the SARS-CoV-2 pandemic, becoming a major global health concern [2]. Psychiatric nosology recognizes stress as the factor that shapes, to various extents, almost all mental health outcomes [3]. The definition of perceived stress refers to the subjects’ feelings and thoughts about the level of stress experienced in response to stressful life events [4]. Perceived stress can vary from person to person based on their unique socio-demographic characteristics, lifestyle, experiences, and coping mechanisms; thus, this construct is quite subjective [5]. High levels of perceived stress may be related to an increased risk of developing disorders such as depression, anxiety, and post-traumatic stress disorder, thus being viewed as a marker of these problems [6,7]. Prolonged or chronic stress can negatively affect not only the brain, but through neuroendocrine dysregulation, it can damage virtually all organs and tissues directly or through functional circuits [8], including disrupting body homeostasis, triggering immune function and changes in hormone levels, inducing an acute phase response commonly associated with infections and tissue damage, and increasing levels of circulating cytokines and various biomarkers of inflammation [9,10,11]. In The International Classification of Diseases 11th Revision (ICD-11), the World Health Organization (WHO) Working Group isolated ‘disorders specifically associated with stress’ as a new group of illnesses, highlighting their substantial detrimental effect on contemporary societies. From this point of view, stress-related disorders could be regarded as those that contribute to the disease burden for both patients and the healthcare system worldwide, being the dominant origin of years lived with disability [12,13,14].

In this context, there is an urgent need to improve pharmacotherapy and psychotherapy [15] and to search for non-pharmacological methods to reverse the increasing trend of mental health deterioration triggered by stress. Changes in lifestyle, including inappropriate eating behaviors, are recommended as a core element of mental health care in clinical practice [14,16]. The growing evidence suggests that dietary choices may be linked to some mental disorders, such as stress, sleep difficulties, and depressive symptoms [17,18], and that improving diet quality can be profitable for mental health [14]. A review by Grajek et al. 2020 identified the rational diet and consumption of psychobiotics and antioxidants, together with physical activity, as potential therapeutic options supporting the management of mental problems [19]. The fact that a separate discipline, nutritional psychiatry, has emerged at the interface of nutrition science, dietetics, and psychology corroborates the usefulness of adhering to a proper diet in maintaining and restoring mental health [20]. Some food ingredients are indicated as particularly beneficial due to their impact on improving the functioning of the immune system, reducing acute or chronic neuroinflammation, and regulating the level of stress hormones [17]. Such ingredients include polyphenols, lignans, and phytosterols [17,21,22]. The pleiotropic effect of lignans and phytosterols results from, among others, their antiviral, antioxidant, and anti-inflammatory properties. Additionally, some of them may exhibit hepatoprotective, immunosuppressive, antiplatelet, cardiovascular, antimutagenic, antiproliferative, antiangiogenic, and hypotensive activity and may also restructure the gut microbiome [22,23]. They exert a broad spectrum of biological activities that may prevent or support the treatment of many diseases [22], and the evidence from observational studies suggests that higher intakes of polyphenols may be inversely associated with the risk of cognitive disorders [24] and other chronic diseases, like cardiovascular diseases [25] and hypertension [26]. Lignans and phytosterols can have an antidepressant effect related to increasing serotonin levels, reducing cortisol levels, suppressing neuroinflammatory reactions, including in microglia, and inhibiting stress-induced damage to intestinal barrier integrity [27,28,29]. A diet rich in these phytochemicals can enhance stress-related oxidative damage, promote mental health, and help reduce the risk of developing mental illness.

The wellness-promoting abilities of lignans and phytosterols suggest that they might be excellent nutritional candidates for supporting the fight against stress and mental disorders. Several studies on animal models have shown the preventive effects of lignans and phytosterols on stress, depression, and behavioral disorders [30,31]. Dietary interventional trials have found that lignans could significantly reduce blood pressure during mental stress induced by a frustrating cognitive task among postmenopausal women with vascular disease [32], and the polyphenol-rich diet could help in the improvement of mental health status and support a reduction in depressive symptoms in the middle-aged hypertensive population [33]. However, nowadays, observational studies linking lignan and phytosterol intake with mental health and its predictors, such as perceived stress, are scarce. Considering the destructive impact of prolonged stress on mental and physical health, particularly during the COVID-19 pandemic, it seems necessary to expand research on searching for nutritional factors effective in the prevention and treatment of negative mental conditions. Therefore, we aimed to explore the association between lignans and phytosterol intake and the level of perceived stress in a group of young individuals without comorbidities during the COVID-19 pandemic in Poland.

## 2. Materials and Methods

### 2.1. Study Population

The study started in 2020 and was conducted in Cracow, a city located in southern Poland. The details and aims of the study have been described elsewhere [34]. In short, it was focused on characteristics of eating habits, body composition, physical activity, the intestinal microbiota, and selected psychological and socio-demographic factors in a sample of young non-obese adults, as well as on testing the relationships between them. The volunteers were recruited through social media and dissemination and promotion requests among familiars [34]. Using the same research method, men and women were examined in July–December 2020 and October–November 2020, respectively. A sample of 100 people was planned for examination to provide a type I error at level 0.05 and power at 0.8. Only individuals adhering to a vegetarian or traditional diet for at least 12 months preceding the measurements and not using antibiotics or probiotics for 3 months before the time of recruitment were selected. The study comprised generally healthy young adults fulfilling the following inclusion criteria: (1) non-obese individuals (body mass index (BMI) < 30 kg/m^2^), (2) individuals aged 25–45 years, and (3) individuals without any chronic disease. Of the total, 104 subjects responded to the invitation. In the first visit, they were carefully educated to maintain a daily routine and not change their diet or physical activity during the next 7 days of examination. The project of the study assumed that it would be a typical week of life, similar to the previous months, accurately reflecting habitual diet and activity patterns. For each subject, the time frame of the examination was selected to not contain any unusual days (i.e., special events such as celebrations). During 7 consecutive days, participants registered their dietary records and physical activity using the Polar M430 (Polar Electro Oy, Kempele, Finland) watch received on the first visit. The participants received careful education on how to keep dietary records for one week of observation using, e.g., kitchen scales or a professional website showing portion sizes of food and the amounts in grams. To facilitate the process of precisely writing down everything eaten during each day of observation, the subjects additionally obtained written examples of properly prepared food diaries. At the same time, to optimize the accuracy of the device measuring the total energy expenditure (TEE), number of steps, and sleep time in accordance with standardized guidelines, the experimenter gave detailed instructions on how to wear the watch. In particular, subjects were asked to place the device on the wrist of the nondominant hand and to take it off only during a bath or shower. After a week, at the second visit, the socio-demographic survey, perceived stress intensity assessment, and bioelectrical impedance analysis were conducted to collect data from each participant. A calibrated segment analyzer (Tanita BC-418 MA, Tokyo, Japan), an 8-electrode device using a constant current source with frequencies of 6.25 kHz, 50 kHz, and 90 μA, was applied to measure body composition and body weight with an accuracy of 0.1% and 0.1 kg, respectively. The procedure was carried out in accordance with the manual; the equipment was placed on a firm, flat surface, and subjects were instructed on how to correctly stand barefoot on the platform for measurement. In subjects without shoes, height was determined with a portable stadiometer (SECA 213, Hamburg, Germany) to the nearest 0.1 cm. Although it is not the subject of the current investigation, stool samples brought by individuals were collected and, within 36 h, delivered to a microbiological laboratory for intestinal microbiota analysis (KyberKompact Pro, Poznan, Poland). The flow diagram presents the details of the selection process (Figure 1).

### 2.2. Tools

The Perceived Stress Scale (PSS) in the Polish-validated version [35] was implemented to measure the perception of stress during the last month before the study. A four-week-recall period length was designed as in the original tool due to the possibility of a decrease in the predictive accuracy of the PSS in the case of a longer time. The instrument consisted of 10 questions, including information about thoughts and feelings that referred to situations appraised as stressful. The responses covered the following options: 0 = never, 1 = almost never, 2 = sometimes, 3 = fairly often, and 4 = very often. Four items (PSS 4, PSS 5, PSS 7, and PSS 8—the content of individual questions can be found in the Appendix A) were obtained by reversing responses according to the rule 0 = 4, 1 = 3, 2 = 2, 3 = 1, 4 = 0, such that the higher score of each question reflected the higher stress. After that, the total score was calculated as the sum of points across all individual items, with the result oscillating between 0 and 40. To facilitate interpretation concerning the whole population, sten scores were applied next, divided into categories: (i) low perception of stress: sten scores of 1–4 (0–13 points), (ii) moderate stress perception: sten scores of 5–6 sten scores (14–19 points), and (iii) high level of perceived stress: sten scores of 7–10 (20–40 points) [30]. In a current study, the median value of PSS (16.5 points) was considered a threshold defining elevated (≥6 stens) and normal perceived stress (<6 stens) groups. All participants were made aware of the research conditions and procedures, and each signed an informed consent document before taking part in the study. The study protocol was approved by the Bioethics Committee of Jagiellonian University (No. 1072.6120.5.2020 and 1072.6120.202.2019) and was performed in accordance with the Declaration of Helsinki for medical research.

### 2.3. Assessment of the Diet

Based on 7-day food diaries delivered by participants, we obtained the energy intake and nutritional content of each product via the Dieta 6.0 program (National Food and Nutrition Institute, Warsaw, Poland), with the built-in latest release of the Polish food composition database. The software is based on the nutrition profile of typical regional recipes and takes into account retention factors, depending on the method of preparing the dishes. The product-specific mean content of lignans and plant sterols was calculated by implementing the database developed by Witkowska et al. covering many types of dishes typically consumed in Poland. Details of the method were explained elsewhere [27,36]. Shortly, the database for plant sterols was created based on 13 data sources with published food composition data for these phytochemicals, such as international food databases (e.g., the British Database of Food Composition, the USDA database), and other scientific data from the literature searched in PubMed, Scopus, Google Scholar, and Web of Science. The database for lignans (total lignan, lariciresinol, matairesinol, pinoresinol, and secoisolariciresinol) used primarily data from the Dutch lignan database and some other literature data sources presenting the content of lignans in oils, nuts, seeds, and beverages. Phytochemical, macronutrient, and micronutrient intake was calculated by multiplying the content of each food by its daily consumption. In the last step for each subject, the summation of all foods recorded in their diary was performed. Intake of lignans in total and subclasses: lariciresinol, matairesinol, pinoresinol, and secoisolariciresinol, and phytosterols in total and subclasses: stigmasterol, campesterol, and β-sitosterol, was analyzed in a current study.

### 2.4. Statistical Analysis

Intake of lignans, plant sterols, and nutrients was corrected for total energy intake using the residual method [37]. When regarded as a quantitative feature, the consumption of each phytochemical was logarithmically transformed (with the Log2 function) to improve the normality of the distribution. The continuous variables were described by means with standard deviations (mean, SD) or medians with quartiles (Me, Q1–Q3), and categorical variables were illustrated by counts with frequencies (*n*, %). Subjects’ background characteristics were presented separately for groups with normal and elevated perceived stress and then compared using the Student’s t-test or chi-squared test for independence. To detect phytochemical consumption, which was linked with perceived stress, and to identify components of PSS that were most strongly connected with diets abundant in phytochemicals, the correlation analysis on logarithmically transformed intakes was performed using Pearson’s method. Because matairesinol was present in the diet of participants in very small amounts and its intake was not correlated with any component of PSS score, we restricted further analyses only to total phytosterols with subclasses: stigmasterol, campesterol, and β-sitosterol, and total lignans and three sub-classes: lariciresinol, pinoresinol, and secoisolariciresinol, omitting matairesinol (see Figure 2 and Appendix A).

As some of the main dietary sources of phytosterols and lignans are common, the analysis for each type of phytochemical was performed separately. Differences in the prevalence of elevated perceived stress between tertiles (T1, T2, and T3) of lignans and phytosterol consumption were checked with a chi-squared test or with univariable and multivariable logistic regression analysis. Additionally, odds ratios (ORs) with 95% confidence intervals (CIs) associated with doubling exposure or with shifting to a higher tertile category of intake were calculated. Taking each multivariable analysis and each phytochemical one at a time, we set three models: (i) Model 1: adjusted to alcohol drinking and smoking status; (ii) Model 2: adjusted to alcohol drinking, smoking status, age, and physical activity; and (iii) Model 3: adjusted to alcohol drinking, smoking status, BMI, sex, body fat, age, and physical activity. The Akaike Information Criterion (AIC) for each model was calculated for comparison of alternative analyses with different covariates. Lower AIC values demonstrate that a given model should be preferred over another in terms of better fitting to the data Therefore, the level of superiority was reflected by the difference in AIC (ΔAIC) between two competing models. A threshold value of 6 was set as a cut-off point, with the rule that the model with ΔAIC ≥ 6 showed small support in the data. Finally, to test whether the phytochemical consumption was related to other dietary variables, a comparison of the distribution of macro- and micronutrients by tertile categories of total lignan intake and total phytosterol intake was conducted by depicting medians and quartiles, followed by performing an ANOVA rank Kruskal-Wallis test. *p*-values from two-sided tests were reported at a significance level of 0.05. R software (Development Core Team, Vienna, Austria, version 4.0.4) was applied for all the statistical analyses.

## 3. Results

### 3.1. Baseline Characteristics of Participants

The study was conducted on 104 people, three-quarters of whom were men and only one-fourth were women. People with PSS lower than 6 stens did not differ from those with PSS higher or equal to 6 stens in basic characteristics such as age, BMI, physical activity, marital status, smoking status, alcohol consumption, and total energy intake. However, a group of persons with elevated PSS contained a more than two-times larger percentage of women (34.6% vs. 15.4%). A comparison of the distributions of selected factors describing the study sample between both premade PSS categories is presented in Table 1.

### 3.2. Correlation Analysis of Lignan and Phytosterol Intake with Total PSS Score and Its Components

We started with correlation analysis, testing pairwise associations of log2-transformed intakes of lignans and phytosterols with total PSS and its components. In the examined sample of young adults, all considered phytochemicals except matairesinol were significantly correlated with at least one question of the scale, thus matairesinol was excluded from further analyses.

Considering only phytochemicals after the aforementioned restriction, all study relationships showed negative correlations of varying strength. Regarding lignans, there is a significant association with total PSS and with items “In the last month, how often have you been upset because of something that happened unexpectedly?” (PSS 1), “In the last month, how often have you felt that you were unable to control the important things in your life?” (PSS 2), reversed “In the last month, how often have you felt that things were going your way?“ (PSS 5) and reversed “In the last month, how often have you felt that you were on top of things?” (PSS 8) was found for total lignans and for each individual subclass except secoisolariciresinol. In turn, consumptions of all phytosterols—in total, stigmasterol, campesterol, and β-sitosterol—were negatively correlated with total PSS and items “In the last month, how often have you been upset because of something that happened unexpectedly?” (PSS 1), reversed “In the last month, how often have you been able to control irritations in your life?” (PSS 7) and “In the last month, how often have you felt difficulties were piling up so high that you could not overcome them?” (PSS 10). Details, including the content of all questions, are presented in Appendix A.

### 3.3. Preliminary Findings—Univariate Analysis of the Association between Lignan and Phytosterol Intake and PSS

Next, we compared the means of lignan and phytosterol intake between subjects with different levels of perceived stress (dichotomized using the median cut-off point). People with normal levels of perceived stress tended to consume more of all considered phytochemicals compared with the elevated perceived stress group; however, the differences were statistically significant only for total phytosterols and all their subclasses but not for lignans (Figure 3 and Appendix A).

To verify a dose-response relationship between individual phytochemicals and high PSS, we considered each exposure in two forms: (i) as logarithmically transformed intake, and (ii) as the variable expressed in tertile categories. A negative linkage of perceived stress with intake of pinoresinol, phytosterols in total, and each specific subclass of phytosterols was shown in a univariable logistic regression analysis for a one-unit increase in log2 of intake. Some evidence of a decreasing trend across tertiles was also observed for all phytosterols, excluding campesterol. Doubling consumption of pinoresinol was associated with diminishing the odds of elevated PSS by about 20% (OR = 0.79, 95% CI: 0.62, 1.00, Table 2).

Likewise, risk reduction accompanying a twice-larger intake of total phytosterols was assessed at a level of 73% (OR = 0.27, 95% CI: 0.09, 0.68), at 48% for stigmasterol (OR = 0.52, 95% CI: 0.28, 0.89), at 60% for campesterol (OR = 0.40, 95% CI: 0.15, 0.97), and at 70% for β-sitosterol (OR = 0.30, 95% CI: 0.11, 0.73, Table 2). The results of the analysis in tertiles generally were in agreement with the aforementioned findings. The frequency of the occurrence of elevated perceived stress differed between three categories of consumers, showing a decreasing trend in a clearly linear form or resembling the reverse J-shape (Table 2).

### 3.4. The Association of High Perceived Stress with the Amount of Phytochemicals Consumed in Diet—Multivariable Analysis

The multivariable analysis confirmed the main aforementioned preliminary findings, which referred to crude estimates. The elevated perceived stress was in a reverse J-shaped relation to both total lignan and pinoresinol intake. Additionally, the elevated value of PSS was negatively associated with dietary intake of total phytosterols, stigmasterol, and β-sitosterol, with evidence of a decreasing trend across tertiles. We obtained stable, robust assessments independently of the covariates included in the particular models (Table 3). After controlling for alcohol drinking, smoking status, BMI, sex, body fat, age, and physical activity, persons from the second tertile of intake compared with those from the first tertile had: 67% lower odds of high PSS in the case of total lignans intake (OR = 0.33, 95% CI: 0.11, 0.97 for T2 vs. T1 and OR = 0.65, 95% CI: 0.22, 1.84 for T3 vs. T1), 78% for lariciresinol intake (OR = 0.22, 95% CI: 0.07, 0.66 for T2 vs. T1 and OR = 0.40, 95% CI: 0.13, 1.15 for T3 vs. T1), and 76% lower odds of high PSS in the case of pinoresinol intake (OR = 0.24, 95% CI: 0.08, 0.70 for T2 vs. T1 and OR = 0.31, 95% CI: 0.10, 0.87 for T3 vs. T1). Among lignans, only pinoresinol intake showed a roughly linear shape of association with PSS. Regarding plant sterols, in all analyses with the main exposure variable incorporated as continuous or divided into tertiles, evidence of protection against the prevalence of perceived stress was noted for total phytosterols, stigmasterol, and β-sitosterol, and the effect was approximately proportionally dependent on dose. Independently of the covariates included in a fully adjusted model, the odds of elevated PSS in the highest tertile compared with the lowest were reduced by about 82%, 72%, and 67%, respectively (OR = 0.18, 95% CI: 0.05; 0.56 for total phytosterols; OR = 0.28, 95% CI: 0.09, 0.85 for stigmasterol; and OR = 0.33, 95% CI: 0.11, 0.96 for β-sitosterol). Simultaneously, a significant decreasing trend across tertiles was found; shifting to a higher tertile category of intake was accompanied by diminishing odds of elevated perceived stress of about 79%, 55%, and 76%, respectively (Table 3). Finally, doubling the intake was associated with significantly lowering the odds of elevated PSS in the case of total lignans, pinoresinol, and total phytosterols and each subclass. The results were visualized in Figure 4 and Appendix A.

The AIC values demonstrated that in all cases, the models with the most supporting data were those in which we controlled only for alcohol consumption and smoking status. All models with added age and physical activity as covariates were equivalent to the corresponding Model 1 (ΔAIC < 6). Only in very few situations, fully adjusted models with additionally incorporated BMI, sex, and body fat showed smaller support in the data (ΔAIC ≥ 6); however, estimates were stable and consistent. In the text and plots, we presented results from fully adjusted models; for details referring to more parsimonious models, we refer to Table 3.

### 3.5. Relationship between Consumption of Phytochemicals and Other Dietary Food Products, Macro- and Micro-Nutrients

Finally, we checked the distribution of macro- and micronutrients by tertiles of total consumption of polyphenols and phytosterols (Table 4). Not surprisingly, a higher intake of both phytochemicals was accompanied by an increased intake of plant protein, dietary fiber, microelements (potassium, iron, copper, and manganese), and β-carotene, but with decreased consumption of total fat. Moreover, a positive association with intake of polyunsaturated fatty acids (PUFA), omega-3 and omega-6 fatty acids, magnesium, vitamin E, and vitamin B6 was observed for phytosterols, whereas vitamin A for lignans. On the contrary, SFA and cholesterol are negatively correlated with phytosterol intake.

## 4. Discussion

The growing evidence suggests that mental stress sequelae, including its effects on mental and physical health, is a common burden in developed countries [38]. Conducting epidemiological research examining the mental health problems in different subpopulations was put forward as a priority, and the search for modifiable factors positively influencing mental health has gained importance, including the period of the COVID-19 pandemic [39]. This is all the more important as the range of pharmacotherapies available for the treatment of psychiatric disorders caused by chronic stress is limited and suboptimal in terms of efficacy and tolerability, and their use is often associated with serious side effects [40,41]. One of the preventive methods that could help counter the impact of stress is healthy, nutrient-dense food containing antioxidants and other compounds essential for proper brain function and mood regulation [17]. Dietary factors can influence the risk of depression through a variety of mechanisms. Several ways in which food can help tame stress have been proposed, including increasing serotonin levels, lowering cortisol and adrenaline, the stress hormones, strengthening the immune system, and lowering blood pressure [32,42]. The use of antioxidant supplements as a strategy for managing stress is a topic of ongoing research and debate [43,44].

Potential candidates meeting these conditions are phytochemicals, which, through regulation of normal brain function and anti-inflammatory, antioxidant, and immunostimulatory effects, influence mood and mental health [45]. Dietary polyphenols have been proven to enhance cognitive ability [24]. Numerous experimental, preclinical, and clinical studies have revealed the therapeutic antidepressant potential of various phytochemicals. Although research into the effects of phytosterols and plant lignans on mood disorders such as depression and stress is still at an early stage, there is some evidence to suggest that they may play a role in reducing the symptoms of these conditions, often similar to antidepressants [21,31]. Therefore, we conducted a study of the association between the level of PSS and the dietary intake of lignans and phytosterols considered to be capable of having a positive effect on mood, stress, and behavior. We examined young, healthy individuals identified as one of the groups at particular risk of stress during the pandemic.

The results of the current study confirmed that specific lignans and phytosterols are associated, to a various extent, with perceived stress. Independently of alcohol drinking, smoking status, BMI, sex, body fat, age, and physical activity, we showed a reverse J-shaped relationship between the odds of high PSS and both total lignan and pinoresinol intake. A negative association of elevated perceived stress with dietary intake of total phytosterols, stigmasterol, and β-sitosterol was also observed, with evidence of a decreasing odds ratio trend across tertiles of intake.

The correlational relationships found in our research demonstrate the potential ability of the analyzed components to protect against stress, revealing that high consumption of total lignans and each subclass, as well as total phytosterols, stigmasterol, campesterol, and β-sitosterol, can be negatively linked with PSS as well as its components, such as feelings of having problems with the management of important things and controlling irritation, nervousness, stress, and the perception of inefficiency in overcoming difficulties.

Concerning lignans, no other studies have investigated such a relationship; however, several experiments in animal models and humans confirm their positive effect on mental health and stress reduction.

Administration of sesamin, the major sesame oil lignan, attenuated behavioral and psychological disorders induced by chronic mild stress in mice, reducing depressive, aversive, repetitive, and anxiety episodes [30]. In a rodent model, supplementation of linseed rich in secoisolariciresinol diglucoside in the maternal diet was associated with some protection against depressive symptoms in the offspring [46]. Secoisolariciresinol in two classical models of depression in ovariectomized mice exhibited antidepressant properties after intragastric administration, substantially reducing the immobility time of mice subjected to intervention compared to controls [47]. Furthermore, a recent clinical study has shown that treatment with sesamin in combination with astaxanthin, an antioxidant belonging to the carotenoid family, can improve mild cognitive impairment in humans [48]. Dietary supplementation with sesamin also promoted the recovery of healthy volunteers from mental fatigue [49], and flaxseed administration significantly lowered blood pressure during mental stress induced by a frustrating cognitive task compared to a baseline diet used before treatment in postmenopausal women with vascular disease [32].

Much hope for the use of plant phytochemicals in reducing stress is provided by the association found in our study between lower levels of PSS and a higher intake of total phytosterols and all their subclasses. Although there are no similar results in this regard in the literature, there is also a growing body of scientific evidence supporting the pharmacological properties of phytosterols, including their human mood-enhancing properties [22], and several studies have shown that dietary intake of phytochemicals is inversely related to the severity of depression and positively related to quality of life. β-sitosterol and its derivatives exerted their antidepressant effects in behavioral despair tests in mice used as animal models, reliably reflecting human depression [31]. Furthermore, β-sitosterol also reversed behavioral deficits and restored working memory in transgenic animals [50]. A study conducted in a group of adolescent girls documented a lower chance of depression (50%) and poor quality of life (38%) in those who were characterized by the highest dietary phytochemical index (DPI) compared to participants in the lowest quartile of DPI [51]. Similar relationships were obtained among female adults; women in the highest DPI tertile showed a lower prevalence of depressive symptoms, anxiety, and psychological distress compared to those in the lowest tertile [52].

The observed association between the likelihood of high PSS and low intake of lignans and pinorezinol, total phytosterols, stigmasterol, and β-sitosterol may be due to the broad spectrum of actions of these compounds, including neuroprotective, anti-inflammatory, and antioxidant effects [21,23]. Supplementation of the extract of sesame cake (SLE) and sesamin inhibited oxidative stress in the brains of depressed mice [53]. Four-week administration of sesamin in mice with diabetic retinopathy resulted in suppression of microglia activation, which plays a role in severe forms of psychiatric disorders [54]. In the mouse microglia cell line BV-2 [55], primary rat microglia cells [56], and neuronal cells of rodents [57], sesamin and/or sesamol exhibited antioxidant and anti-inflammatory activities [54]. Sesamin administration reduced levels of tumor necrosis factor-α (TNF-α) and intercellular adhesion molecule-1 (ICAM-1) and suppressed inducible nitric oxide synthase (iNOS) expression in a mouse model of diabetic retinopathy [54]. Sesamin treatment also substantially diminished inflammation and oxidative stress in a mouse model of ischemic stroke [58] and in traumatic brain injury [59]. Sesamin administration lowered concentrations of lipopolysaccharide (LPS) in circulation, suppressed neuroinflammatory responses [30], and also inhibited depressive behavior, anxiety, and memory loss by reducing neuroinflammation induced by chronic mild stress in mice [60]. Antioxidant and anti-inflammatory effects were also demonstrated by pinorezinol treatment in a mouse model of Alzheimer’s disease [61] and in a middle cerebral artery occlusion model, resulting in reduced brain damage [62]. In vivo and in vitro studies showed that pinorezinol-4-O-β-D-glucopyranoside isolated from plums exhibited antioxidant activity [63]. One proposed mechanism by which lignans may affect mood is their ability to promote the production of polyunsaturated fatty acids such as EPA and DHA. These fatty acids have been shown to have anti-inflammatory properties and may help counteract pro-inflammatory cytokines that are produced during periods of stress and depression [61,64]. The anti-stress effect of phytosterols is due to their ability to cross the blood-brain barrier, where they are irreversibly accumulated and incorporated into cell membranes, thus playing a positive role in reducing inflammation [65]. The antioxidant effects of phytosterols are linked to their high capacity to scavenge free radicals, stabilize cell membranes, and enhance antioxidant enzymes [22]. The therapeutic potential of phytosterols against oxidative stress and inflammation is confirmed by numerous studies. A milk-based fruit drink enriched with phytosterols blocked oxidative stress and reduced interleukin (IL)-8 and IL-6 levels in differentiated Caco-2 cells [66] and exhibited anti-inflammatory activities in an experimental mouse model of chronic ulcerative colitis [67]. In another mouse model of colitis, phytosterol administration decreased oxidative stress in addition to its anti-inflammatory effects [68]. Studies in apolipoprotein-deficient mice showed that phytosterol-enriched diets were strongly associated with higher production of anti-inflammatory cytokines and lower production of pro-inflammatory cytokines [69].

In human studies, the administration of plant sterol and sterolin capsules to ultramarathon participants resulted in a lower inflammatory response and a reduction in subtle immunosuppression induced by excessive physical stress [70]. Individual phytosterols showed similar effects. Incorporation of β-sitosterol into the membrane of HT22 cells and cultured hippocampal cells inhibited oxidative stress mediated by glucose oxidase and lipid peroxidation [71]. β-sitosterol also displayed antioxidant potential by scavenging free radicals of different natures both in vitro and in vivo in an Alzheimer’s disease model [50]. Stigmasterol exerted anti-inflammatory and antioxidant effects in a dose-dependent manner in the BEAS-2B cell model and in vivo in asthmatic mice [72]. In in vitro studies using human neuronal cells, pretreatment with stigmasterol maintained reactive oxygen species (ROS) levels in cells and prevented cell death induced by oxidative stress. In addition, sirtuin 1 (SIRT1) activity was stimulated by stigmasterol, and the result was comparable to resveratrol [73]. A protective effect of lignans against the effects of chronic stress-induced anxiety disorders resulting from modulating dopamine, norepinephrine, and serotonin levels, c-Fos expression, and corticosterone levels, as well as inhibiting the enzymatic activity of MAO-A monoxidase responsible for the deamination of serotonin, adrenaline, and norepinephrine dopamine and tyramine, and MAO-B monoxidase showing affinity for dopamine and serotonin, has also been observed in animal models [60,74,75]. In mouse models of depression and/or chronic stress, SLE and sesamol supplementation improved serotonin levels in the brains [53]. Administration of secoisolariciresinol significantly increased monoamine (norepinephrine and dopamine) levels [47], as well as stopped and even reversed the excessive release of corticosterone and adrenocorticotropic hormones [76]. Lignan intake in postmenopausal women under stressful conditions was able to reduce plasma cortisol concentrations [32]. This anti-anxiety effect of lignans may be related to their anti-stress effects through modulation of the hypothalamic-pituitary-adrenal (HPA) axis, which plays an important role in both adaptation and stress response [74]. The anti-inflammatory effects of polyphenols, such as lignans, on microglia may have implications for neuroprotection and the prevention of neurodegenerative diseases [17]. Some studies also suggest that lignans and phytosterols may promote neurogenesis, the growth and development of new neurons in the hippocampus, a brain region involved in learning and memory, and increase synaptic plasticity, the ability of synapses to strengthen or weaken over time in response to changes in neuronal activity [65]. Sesamin may also prevent disruption of the blood-brain barrier, which was shown in a mouse model of traumatic brain injury [77]. Sesamin treatment also inhibited stress-induced damage to intestinal barrier integrity and restructured the gut microbiome [30], whose composition is strongly correlated with behavioral disturbances, serotonin, norepinephrine, and LPS levels [78]. Interestingly, the anti-anxiety, sedation-inducing, and muscle-relaxing effects of some of the lignans and phytosterols are similar to those of other antidepressants and sedatives [53,79,80,81]. Also, the frequency of depression and mood disorder symptoms during the SARS-CoV-2 pandemic tended to be greater among participants with pre-existing poor physical health compared to the general population [82]. In this context, the pleiotropic activities of phytosterols and lignans and their efficacy in disease management through, among others, chemopreventive, antidiabetic, antiatherosclerotic, cardioprotective, antiplatelet, and antimicrobial properties [22,79] are important in reducing the tragic consequences of SARS-CoV-2 virus infection. In addition, the confirmed antiviral efficacy of polyphenols, including lignans and plant sterols, against the SARS-CoV-2 virus, resulting from their ability to bind to protein peak sites on the ACE2 receptor used by SARS-CoV-2 to infect cells, predisposes these compounds to wide use in prevention and support of treatment [27].

It is not surprising that a higher intake of the phytochemicals analyzed was accompanied by an increased intake of plant protein, dietary fiber, micronutrients (potassium, magnesium, iron, and copper), and β-carotene because excellent sources of both lignans and phytosterols are oilseeds, whole-grain cereals, nuts, almonds, pumpkin seeds, wheat germ, legumes, various vegetables and fruits, as well as coffee, tea, and wine [31,36,81,83].

The observed positive association of phytosterols with intakes of PUFAs, omega-3 and omega-6 fatty acids, and manganese is due to the high content of fats rich in mono- and polyunsaturated fatty acids, including n-3, and the low content of SFAs and cholesterol in products that are dietary sources of phytosterols (nuts, pistachios, and flaxseed). The association between a larger intake of the phytochemicals analyzed and a reduced intake of total fat is intriguing. Witkowska et al. showed that in the traditional diet of a Pole, cereals and fats provide 61% of all plant sterols, while vegetables and fruit provide only 19% [36].

Similarly, in our study, the diets of those with the lowest levels of lignans and phytosterols were mainly derived from cereals and vegetable fats. In addition, these diets were characterized by the highest levels of fat, saturated fatty acids, animal protein, cholesterol, and sucrose and the lowest levels of fiber, i.e., they were similar to Western-type diets showing pro-inflammatory effects [84,85]. Our analysis revealed that the dietary matrix of the individuals with the highest phytochemical content in our study consisted mainly of fruit, vegetables, and cereal products. The main sources of lignans were whole grain cereals, buckwheat groats, broccoli, carrots, potatoes, and legumes, while plant sterols were provided by oils and oilseeds. In the diets of the volunteers we studied, stigmasterol came mainly from soy products, and campesterol, in addition to rapeseed oil, was provided by bananas, onions, potatoes, and oatmeal. However, this was not a typically plant-based diet, given the fact that vitamin A was positively correlated with total lignan intake—lignans also came from animal products (dairy products, meat, eggs, and fish) [68]. This dietary pattern is close to the Mediterranean dietary pattern, rich in fruit and vegetables, and characterized by a high content of bioactive compounds, including lignans and phytosterols. Adherence to a Mediterranean-type diet containing high amounts of fruit, vegetables, legumes, whole grains, nuts, and seeds rich in omega-3 polyunsaturated fatty acids and limiting the intake of processed foods, bakery products, and sweets is a promising treatment strategy for reducing the risk and symptoms of depression [86,87]. This was confirmed by a large meta-analysis conducted in children and adolescents—healthy dietary patterns and higher intake of a healthy diet were significantly associated with fewer depressive symptoms [88]. Adherence to a Mediterranean diet rich in bioactive compounds, including lignans and phytosterols, was also inversely associated with the incidence of depression in adults [89,90], and in those aged 65 years and older, it was associated with a reduction in new-onset depressive symptoms [91].

It therefore appears that altering dietary patterns and diet composition as a modifiable risk factor for depression may be an excellent strategy to support the body’s antioxidant defense mechanisms, counteract the inflammation associated with the onset and severity of mental disorders, and thereby improve public health. While the research in this area is still evolving, there are some potential applications of phytosterols and lignans in public health and nutrition related to mental well-being. The development of a health policy that includes the introduction of a diet rich in lignans and phytosterols should become a key target for early intervention and supportive treatment of mental disorders at any age. As adherence to lifestyle changes is extremely challenging, the implementation of a strategy at the national level is needed.

The current study has some limitations. First, the functioning of people who experienced stress could have been directly and indirectly affected by the timing of the research coinciding with the COVID-19 pandemic. Second, due to economic restrictions, we had to stop the use of face-to-face interviews during the second phase of the study, which resulted in a predominance of men among participants, and we did not conduct longitudinal observation of eating habits and stress. The subjects kept a food diary for a week, and then they completed the PSS questionnaire, which only allowed us to test the hypothesis in one direction, whether nutrition could affect stress but not the other way around, whether stress might be a predictor of nutrition quality. Moreover, the research was observational, so we did not demonstrate causality, and a partial confounding by factors not incorporated into models is possible. Third, the PSS 10 scale has its weaknesses, such as a short list of items, one-dimensionality, and content items that are open, direct, and “not protected” against the effects of the desire to present oneself in a specific light. Moreover, the questionnaire assessed stressor exposure over the last month, so it took a longer time compared to the observation of the participants’ diet.

However, the study has many strengths as well. First, we conducted a detailed nutritional interview reflecting the habitual diet of participants in the time preceding the self-assessment of stress, and we collected broad information about the phytochemical content of consumed products using 16 data sources available in the literature. This enabled us to calculate not only the intake of total lignans and phytosterols but also their specific representatives. Secondly, we performed a comprehensive analysis, taking into account different models with a variety of possible confounders. Third, we applied the PSS 10 questionnaire, and even though it is not a perfect tool, it has numerous advantages, such as good psychometric properties, validation of the Polish language and conditions, wide use in many countries, self-reporting, and the economy of filling out the questionnaire by participants. Finally, similar results in the literature verifying the relationships between habitual intake of lignans and phytosterols and global perceptions of stress are scarce.

## 5. Conclusions

It appears that both lignans and phytosterols, through their pleiotropic effects (antioxidant, anti-inflammatory, or immunostimulating), may support the prevention or reduction of stress and can also play a beneficial role in managing depressed mood states in young people. This observed effect of phytochemicals, combined with their antiviral efficacy, acts as a prophylactic and lowers the chances of mental deterioration in emergencies similar to those observed during the COVID-19 pandemic. A dietary matrix based on a large amount of vegetables, fruits, cereal products, nuts, and oilseeds containing large amounts of lignans and plant sterols and with a small amount of animal products, similar to the Mediterranean dietary pattern, might be an excellent alternative to conventional antidepressants, avoiding the adverse side effects of the drugs. Further research involving a larger sample of women is recommended, allowing for sex-strata analysis and taking into account the diversification of tools examining experienced stress (e.g., in the form of physiological data or a list of stressful events) as well as factors that may be important for the strength of perceived stress (e.g., personality or temperamental properties of the subject, social support). Exploring the longitudinal impact of a diet abundant in lignans and plant sterols on perceived stress could better clarify a temporal association.

## Figures and Tables

**Figure 1 nutrients-16-00445-f001:**
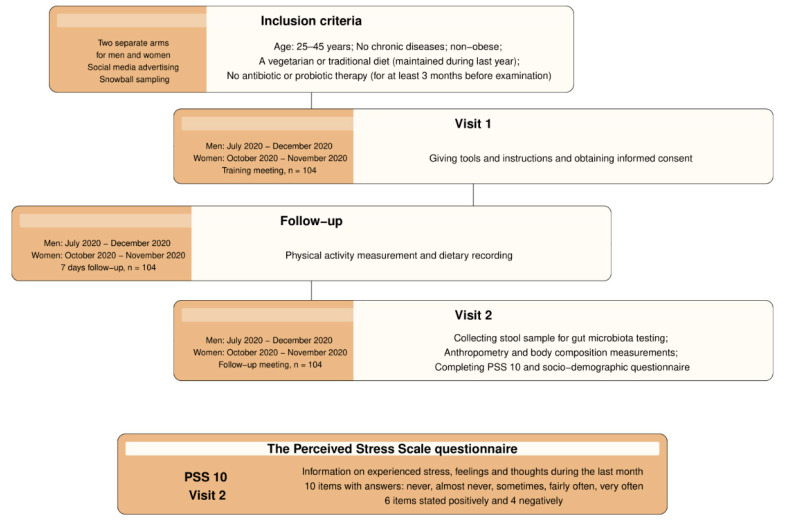
Outline of the research protocol.

**Figure 2 nutrients-16-00445-f002:**
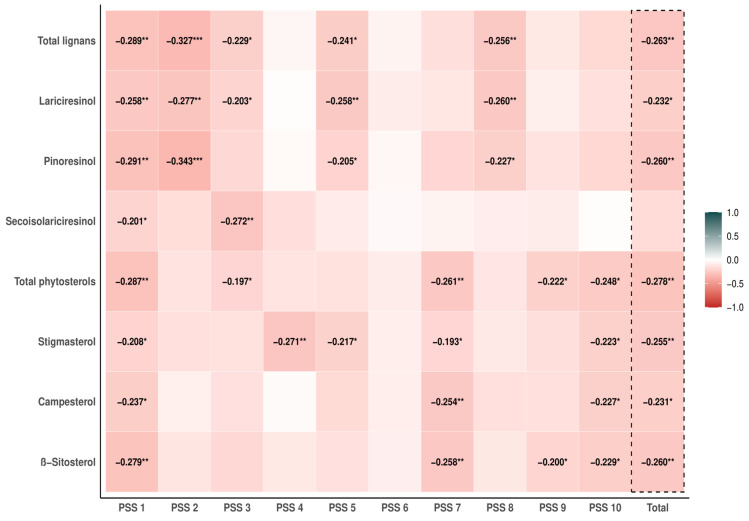
The Pearson correlation coefficients between the Perceived Stress Scale (PSS) and individual items and intake of phytochemicals (*n* = 104). Only statistically significant results are presented. * *p* < 0.05; ** *p* < 0.01; *** *p* < 0.001.

**Figure 3 nutrients-16-00445-f003:**
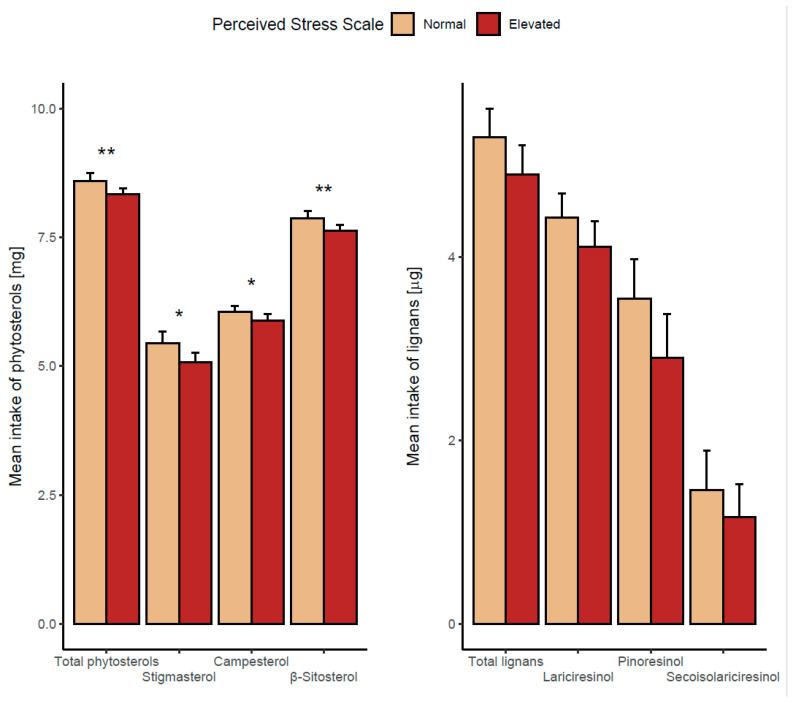
Comparison of the distribution of logarithmically transformed daily intake of specific phytochemicals between respondents with normal and elevated perceived stress. Specific phytochemical intakes were presented in separate plots as expressed in different units. * *p* < 0.05, ** *p* < 0.01 for Student’s *t*-test analysis (*n* = 104).

**Figure 4 nutrients-16-00445-f004:**
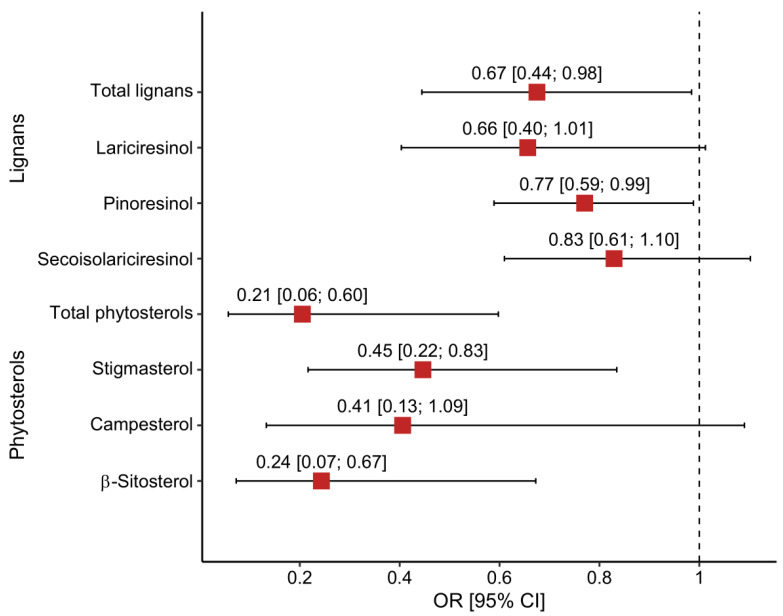
Odds ratios of elevated perceived stress associated with a double increment in phytochemical intake. Results of multiple logistic regression analysis for fully adjusted models after controlling for alcohol drinking, smoking status, BMI, sex, body fat, age, and physical activity. The red squares and horizontal lines represent the phytosterol-specific ORs and 95% CIs (*n* = 104).

**Table 1 nutrients-16-00445-t001:** Baseline characteristics of the study participants (*n* = 104).

Variable	Category/Unit	Total, *n* = 104	Normal Perceived Stress < 6 stens, *n* = 52	Elevated Perceived Stress ≥ 6 stens, *n* = 52
Age ^$^	Years	34.6 (5.7)	34.4 (5.8)	34.7 (5.6)
Sex ^&^	Men	78 (75.0)	44 (84.6)	34 (65.4) *
	Women	26 (25.0)	8 (15.4)	18 (34.6)
BMI ^$^	Kg/m^2^	23.2 (2.6)	23.5 (2.8)	22.9 (2.4)
Log2 PA ^$^	H/d	0.6 (0.2)	0.6 (0.2)	0.6 (0.2)
TEE ^$^	Kcal/d	2535 (456)	2598 (402)	2472 (500)
Sleep duration ^$^	H/d	7.4 (0.9)	7.4 (0.8)	7.4 (0.9)
Energy intake ^$^	Kcal/d	2206 (525)	2259 (475)	2153 (570)
Marital status ^&^	Single/divorced	55 (52.9)	26 (50.0)	29 (55.8)
	Married/cohabiting	49 (47.1)	26 (50.0)	23 (44.2)
BMI [kg/m^2^] ^&^	Normal body weight	74 (71.2)	36 (69.2)	38 (73.1)
	Overweight	30 (28.8)	16 (30.8)	14 (26.9)
Diet ^&^	Traditional	58 (55.8)	25 (48.1)	33 (63.5)
	Vegetarian	46 (44.2)	27 (51.9)	19 (36.5)
BF [%] ^&^	Underweight	11 (10.6)	6 (11.5)	5 (9.6)
	Normal	77 (74.0)	38 (73.1)	39 (75.0)
	Overweight	16 (15.4)	8 (15.4)	8 (15.4)
Smoking ^&^	No	91 (87.5)	48 (92.3)	43 (82.7)
	Yes	13 (12.5)	4 (7.7)	9 (17.3)
Alcohol ^&^	None or moderate	65 (62.5)	34 (65.4)	31 (59.6)
	Regular	39 (37.5)	18 (34.6)	21 (40.4)

* Results are expressed as ^$^ mean (SD) or ^&^
*n* (%), TEE—total energy expenditure, BMI—body mass index, BF—body fat, PA—physical activity, SD—standard deviation; *p* values are based on the chi-squared test of independence or the Student’s *t*-test; * *p* < 0.05 (*n* = 104).

**Table 2 nutrients-16-00445-t002:** Association between phytochemical intake and level of PSS—comparison of distribution of prevalence of high stress by tertiles of consumptions and crude logistic regression analysis for continuous exposure level (*n* = 104).

Daily Phytochemical Intake	Phytochemical-Specific Tertiles			OR (95% CI) of Elevated PSS	
	T1 (*n* = 35)	T2 (*n* = 34)	T3 (*n* = 35)	Per 1 Category of Tertile Increase	Per 1 Unit Increase in Log2
	*n* (%) Elevated Perceived Stress				
Total lignans [μg]	22 (62.9)	12 (35.3)	18 (51.4)	0.79 (0.49; 1.27)	0.72 (0.50; 1.02)
Lariciresinol [μg]	21 (60.0)	13 (38.2)	18 (51.4)	0.84 (0.52; 1.35)	0.72 (0.47; 1.07)
Pinoresinol [μg]	24 (68.6)	12 (35.3)	16 (45.7) *	0.63 (0.38; 1.01)	0.79 (0.62; 1.00) *
Secoisolariciresinol [μg]	18 (51.4)	16 (47.1)	18 (51.4)	1.00 (0.62; 1.60)	0.87 (0.65; 1.13)
Total phytosterols [mg]	24 (68.6)	15 (44.1)	13 (37.1) *	0.52 (0.31; 0.85) **	0.27 (0.09; 0.68) **
Stigmasterol [mg]	23 (65.7)	15 (44.1)	14 (40.0)	0.59 (0.36; 0.95) *	0.52 (0.28; 0.89) *
Campesterol [mg]	20 (57.1)	17 (50.0)	15 (42.9)	0.75 (0.46; 1.20)	0.40 (0.15; 0.97) *
β-sitosterol [mg]	24 (68.6)	12 (35.3)	16 (45.7) **	0.63 (0.38; 1.01)	0.30 (0.11; 0.73) *

T1, T2, and T3—phytochemical-specific tertile groups. Results are expressed as *n* (%), or OR (95% CI), OR—odds ratio, CI—confidence interval, * *p* < 0.05, ** *p* < 0.01 from chi-squared test or logistic regression analysis.

**Table 3 nutrients-16-00445-t003:** Association between phytochemical intake and high perceived stress. Results of multiple logistic regression analysis for tertiles of consumption and for continuous exposure (*n* = 104).

Daily Phytochemical Intake	Phytochemical-Specific Tertiles			Δ AIC	Per 1 Category of Tertile Increase	Δ AIC	Per 1 Unit Increase in Log2	Δ AIC
	T1 (*n* = 35)	T2 (*n* = 34)	T3 (*n* = 35)					
Total lignans [μg]								
Model 1	1 (ref.)	0.31 (0.11; 0.82) *	0.59 (0.22; 1.55)	0	0.77 (0.47; 1.24)	0	0.72 (0.49; 1.02)	0
Model 2	1 (ref.)	0.31 (0.11; 0.83) *	0.58 (0.22; 1.54)	3.9	0.77 (0.47; 1.24)	3.8	0.72 (0.49; 1.02)	3.8
Model 3	1 (ref.)	0.35 (0.12; 1.01)	0.53 (0.19; 1.48)	8.2	0.73 (0.43; 1.22)	5.9	0.67 (0.44; 0.98) *	5.9
Lariciresinol [μg]								
Model 1	1 (ref.)	0.41 (0.15; 1.08)	0.70 (0.26; 1.83)	0	0.84 (0.51; 1.35)	0	0.72 (0.47; 1.08)	0
Model 2	1 (ref.)	0.40 (0.15; 1.07)	0.68 (0.25; 1.81)	3.6	0.83 (0.51; 1.35)	3.8	0.72 (0.46; 1.07)	3.8
Model 3	1 (ref.)	0.48 (0.17; 1.37)	0.64 (0.23; 1.77)	7.4	0.80 (0.48; 1.34)	5.7	0.66 (0.40; 1.01)	5.7
Pinoresinol [μg]								
Model 1	1 (ref.)	0.26 (0.09; 0.69) **	0.39 (0.14; 1.03)	0	0.63 (0.38; 1.02)	0	0.79 (0.62; 1.00)	0
Model 2	1 (ref.)	0.26 (0.09; 0.70) **	0.39 (0.14; 1.04)	3.9	0.63 (0.38; 1.03)	3.8	0.79 (0.61; 1.00) *	3.8
Model 3	1 (ref.)	0.30 (0.10; 0.86) *	0.38 (0.13; 1.07)	7.8	0.61 (0.36; 1.03)	6.2	0.77 (0.59; 0.99) *	6.2
Secoisolariciresinol [μg]								
Model 1	1 (ref.)	0.80 (0.30; 2.11)	0.94 (0.36; 2.45)	0	0.97 (0.60; 1.56)	0	0.84 (0.63; 1.11)	0
Model 2	1 (ref.)	0.78 (0.29; 2.10)	0.94 (0.35; 2.49)	3.6	0.97 (0.59; 1.58)	3.8	0.84 (0.63; 1.11)	3.8
Model 3	1 (ref.)	0.88 (0.31; 2.53)	0.90 (0.32; 2.51)	6.3	0.95 (0.57; 1.59)	5.9	0.83 (0.61; 1.10)	5.9
Total phytosterols [mg]								
Model 1	1 (ref.)	0.33 (0.12; 0.90) *	0.24 (0.08; 0.66) **	0	0.49 (0.29; 0.81) **	0	0.25 (0.08; 0.65) **	0
Model 2	1 (ref.)	0.33 (0.11; 0.90) *	0.24 (0.08; 0.66) **	3.9	0.49 (0.29; 0.82) **	3.9	0.25 (0.08; 0.65) **	3.9
Model 3	1 (ref.)	0.30 (0.10; 0.89) *	0.18 (0.05; 0.56) **	3.6	0.42 (0.22; 0.74) **	5.2	0.21 (0.06; 0.60) **	5.2
Stigmasterol [mg]								
Model 1	1 (ref.)	0.44 (0.16; 1.18)	0.31 (0.11; 0.84) *	0	0.55 (0.33; 0.91) *	0	0.50 (0.26; 0.87) *	0
Model 2	1 (ref.)	0.43 (0.15; 1.21)	0.31 (0.10; 0.85) *	4	0.56 (0.32; 0.92) *	4	0.50 (0.26; 0.88) *	4
Model 3	1 (ref.)	0.35 (0.11; 1.06)	0.28 (0.09; 0.85) *	6.9	0.54 (0.30; 0.93) *	6.8	0.45 (0.22; 0.83) *	6.8
Campesterol [mg]								
Model 1	1 (ref.)	0.73 (0.28; 1.91)	0.55 (0.20; 1.46)	0	0.74 (0.45; 1.21)	0	0.40 (0.14; 0.99) *	0
Model 2	1 (ref.)	0.74 (0.28; 1.94)	0.55 (0.20; 1.46)	3.9	0.74 (0.45; 1.21)	3.9	0.40 (0.14; 1.00) *	3.9
Model 3	1 (ref.)	0.72 (0.25; 2.01)	0.55 (0.19; 1.55)	6.7	0.74 (0.43; 1.25)	6.8	0.41 (0.13; 1.09)	6.8
β-Sitosterol [mg]								
Model 1	1 (ref.)	0.25 (0.09; 0.66) **	0.37 (0.13; 1.00)	0	0.61 (0.37; 1.00)	0	0.29 (0.10; 0.73) *	0
Model 2	1 (ref.)	0.23 (0.08; 0.65) **	0.36 (0.13; 0.98) *	3.8	0.61 (0.37; 1.00)	3.9	0.29 (0.10; 0.73) *	3.9
Model 3	1 (ref.)	0.23 (0.07; 0.67) **	0.33 (0.11; 0.96) *	6	0.58 (0.33; 0.99) *	5.8	0.24 (0.07; 0.67) *	5.8

T1, T2, and T3—phytochemical-specific tertile groups. Results are presented as ORs (95% CIs), * *p* < 0.05, ** *p* < 0.01 from multiple logistic regression analysis, Model 1: adjusted to alcohol drinking and smoking status; Model 2: adjusted to alcohol drinking, smoking status, age, and physical activity; Model 3: adjusted to alcohol drinking, smoking status, BMI, sex, body fat, age, and physical activity.

**Table 4 nutrients-16-00445-t004:** Daily intake of selected micro- and macronutrients and food products (energy-adjusted) by tertiles of lignan and phytosterol intake (*n* = 104).

Daily Intake	Total Lignan Tertiles			Total Phytosterol Tertiles		
	T1 (*n* = 35)	T2 (*n* = 34)	T3 (*n* = 35)	T1 (*n* = 35)	T2 (*n* = 34)	T3 (*n* = 35)
Energy [kcal]	2196 (1886–2468)	2317 (2034–2473)	2145 (1802–2309)	2196 (1925–2514)	2217 (1923–2482)	2161 (1829–2334)
Total protein [g]	88 (74–99)	85 (74–95)	82 (76–95)	85 (79–94)	92 (78–101)	82 (70–93)
Animal protein [g]	51 (30–68)	45 (21–58)	45 (24–59)	52 (43–59)	52 (28–67)	22 (9–45) ***
Plant protein [g]	34 (30–41)	44 (35–53)	40 (32–51) *	32 (28–40)	35 (31–42)	51 (45–60) ***
Arginine [mg]	4526 (3844–5290)	4520 (3874–5140)	4438 (4108–5354)	4171 (3823–5008)	4714 (3973–5513)	4716 (4023–5236)
Fat [g]	77 (69–83)	68 (64–77)	73 (62–76) *	75 (67–78)	76 (69–83)	67 (58–76) *
SFA [%]	27 (22–31)	24 (18–27)	24 (20–29)	29 (24–30)	26 (21–31)	20 (16–25) ***
MUFA [%]	29 (27–34)	27 (23–32)	27 (25–30)	27 (25–30)	29 (26–34)	27 (21–32)
PUFA [%]	13 (11–16)	13 (11–16)	13 (11–17)	12 (9–13)	13 (11–17)	14 (13–20) ***
Omega- FA [g]	2 (1–2)	2 (1–2)	1 (1–2)	1 (1–2)	2 (1–3)	2 (1–3) **
Omega-6 FA [g]	8 (7–12)	8 (7–11)	8 (7–11)	7 (6–8)	8 (7–13)	10 (8–12) **
Total carbohydrates [g]	286 (267–302)	300 (274–330)	300 (281–331)	289 (275–307)	285 (261–327)	307 (280–345) *
Saccharose [g]	43 (34–56)	45 (28–61)	37 (30–47)	47 (34–61)	40 (28–53)	42 (30–48)
Dietary fiber [g]	23 (19–26)	27 (22–35)	29 (26–34) ***	24 (18–26)	25 (21–30)	34 (27–43) ***
Cholesterol [mg]	285 (197–343)	260 (151–374)	274 (192–366)	297 (245–360)	318 (229–460)	191 (80–286) ***
Sodium [mg]	3200 (2951–4019)	3235 (2669–3931)	3527 (3034–4053)	3305 (3111–4127)	3432 (3076–4033)	3069 (2360–3643)
Potassium [mg]	3194 (2892–3772)	3800 (3405–4215)	3763 (3331–4606) **	3245 (2894–3438)	3767 (3297–4207)	4099 (3463–4717) ***
Calcium [mg]	818 (599–969)	875 (751–1044)	809 (734–1010)	847 (750–992)	894 (699–1009)	790 (659–988)
Magnesium [mg]	389 (341–439)	469 (391–534)	426 (378–531) *	376 (299–414)	423 (371–493)	529 (437–595) ***
Iron [mg]	15 (12–16)	16 (14–19)	16 (14–19) **	14 (12–15)	15 (14–17)	17 (16–20) ***
Zinc [mg]	12 (9–14)	12 (11–14)	13 (11–14)	12 (10–14)	13 (11–14)	13 (11–14)
Copper [mg]	2 (1–2)	2 (2–2)	2 (1–2) **	1 (1–2)	2 (2–2)	2 (2–3) ***
Manganese [mg]	6 (5–7)	7 (6–10)	7 (5–8)	5 (4–7)	6 (5–8)	9 (6–11) ***
Vitamin A [µg]	964 (703–1200)	1110 (801–1508)	1383 (1034–1667) **	1041 (876–1376)	1183 (876–1515)	1101 (863–1648)
β-carotene [µg]	3332 (2517–4078)	4683 (2962–6489)	5577 (3660–7568) *	3576 (2364–5321)	4188 (2740–6136)	4825 (3442–8122) *
Vitamin E [mg]	12 (9–14)	13 (10–16)	12 (10–16)	11 (8–14)	12 (10–15)	15 (12–17) **
Thiamin [mg]	1 (1–1)	1 (1–2)	1 (1–2)	1 (1–2)	1 (1–1)	1 (1–2) *
Riboflavin [mg]	2 (1–2)	2 (2–2)	2 (2–2)	2 (1–2)	2 (2–2)	2 (2–2)
Niacin [mg]	20 (16–22)	19 (17–23)	19 (15–24)	19 (17–21)	21 (17–26)	19 (16–23)
Vitamin B6 [mg]	2 (2–2)	2 (2–3)	2 (2–2)	2 (2–2)	2 (2–3)	2 (2–3) **
Vitamin B12 [µg]	6 (0–20)	3 (0–11)	10 (1–23)	2 (1–12)	10 (0–20)	8 (1–16)
Vitamin C [mg]	104 (52–151)	119 (86–153)	135 (90–197)	109 (77–135)	96 (64–157)	139 (109–201) *
Vitamin D [µg]	4 (2–7)	4 (3–10)	4 (3–15)	4 (2–6)	6 (3–15)	3 (2–7)
Alcohol [g]	5 (0–13)	6 (2–17)	6 (2–11)	5 (1–13)	6 (1–11)	6 (2–18)

T1, T2, and T3—phytochemical-specific tertile groups. Results are expressed as Me (Q1-Q3), Me—median, Q1 and Q3—lower and upper quartiles, * *p* < 0.05, ** *p* < 0.01, *** *p* < 0.001 from ANOVA rank Kruskal-Wallis test, FA—fatty acids, MUFA—monounsaturated fatty acids, PUFA—polyunsaturated fatty acids, SFA—saturated fatty acids.

## Data Availability

The data presented in this study are not publicly available due to confidentiality reasons. These data are available on request from the corresponding author.

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
