# Peer review of "Negative Association of Lignan and Phytosterol Intake with Stress Perception during the COVID-19 Pandemic—A Polish Study on Young Adults"

_nutrients, 2024, doi:10.3390/nu16030445_

Round 1

Reviewer 1 Report

Comments and Suggestions for Authors

Dear authors,

Thank you for giving me the opportunity to review your manuscript which, in an extended paper, studies the association between lignan and phytosterol consumption and perceived stress during the COVID-19 pandemic.

The study is very well documented and it is true that the number of participants (over 100) ensures the correctness of the statistical processing but, to strengthen the "truth" value of the results obtained, it would have been better if the number of participants was higher..

The fact that you use the same structure of the manuscript as in reference 26 is sometimes confusing.

Also, the fact that you collected information from men at a different time than women raises a question mark. In May, when you started collecting information from men, the pandemic was at its beginning, there were many unknowns about its evolution, and we suspect that the stress level was higher. In November, when you started the study on women, more information was already known about the period of COVID-19, and the volunteers should have been more familiar with the problems caused by the pandemic.

The fact that, when you describe the chapter MATERIALS AND METHODS, you do not talk about this chapter but refer to reference 29, makes it difficult to follow the study. The same happens in the dietary assessment chapter when you refer to references 26 and 31.

Please describe the above-mentioned chapters in more detail, remembering that this manuscript is the result of a larger study that took shape in several articles.

Also, please explain, in more detail, why you used these inclusion criteria and not others.

In table 1, the average for the duration of sleep($) for all participants is not correctly expressed (0.3 H/d ????)

Many questions are included in the title of figure 2. These could only be written in manuscript without crowding the title of figure 2.

In the manuscript you referred to the attached supplementary materials, but I could not find them. Only the current manuscript is attached.

I appreciate that the limitations of the article are well presented

Good luck!

Author Response

Dear Editor,

We would like to express our appreciation to the reviewers and editorial board for taking the time and effort dedicated to review our manuscript. The reviewers’ comments were very useful and provided insightful guidelines that helped us to improve the scientific quality of the study. We have carefully considered your suggestions. Here we submit a point-by-point response to the questions. All responses for reviewers are in red and main changes to manuscript are highlighted in the manuscript text.

We paste the answers both, here in the designated window below, as well as in the file attached.

RESPONSE TO REVIEWER 1 COMMENTS

Thank you for giving me the opportunity to review your manuscript which, in an extended paper, studies the association between lignan and phytosterol consumption and perceived stress during the COVID-19 pandemic. The study is very well documented and it is true that the number of participants (over 100) ensures the correctness of the statistical processing but, to strengthen the "truth" value of the results obtained, it would have been better if the number of participants was higher..

Author’s response: Dear reviewer, we sincerely appreciate your valuable time and feedback on our article. We agree that it would have been better if the number of participants was higher, however the narrow time window of data collection was equally important for us. In the light of various COVID-19 restrictions, which were obstructing and delaying the examination process, we couldn’t continue the study much longer, so we finished it after obtaining enough information to test our hypotheses.

The fact that you use the same structure of the manuscript as in reference 26 is sometimes confusing.

Author’s response: In reference 27 (previously 26 ) in the same population we tested the association between lignan and phytosterol intake and COVID-19 risk and immune-stimulating microbiota.  In the parts referring to COVID-19 risk reference 27 can have similar structure to the current article due to the fact that we have used exactly the same main explanatory variables and the same method of analysis based on logistic regression. In both publications we would like to apply standard categorisation of tested continuous variables so some level of similarity was inevitable. The division based on quantiles was chosen as one of the most commonly encountered approaches in the literature in a case of lack of alternative widely recognized recommendations for cut-off points. Three groups were determined as the most reasonable option to follow dose-response effect taking into account relatively small sample size.

Also, the fact that you collected information from men at a different time than women raises a question mark. In May, when you started collecting information from men, the pandemic was at its beginning, there were many unknowns about its evolution, and we suspect that the stress level was higher. In November, when you started the study on women, more information was already known about the period of COVID-19, and the volunteers should have been more familiar with the problems caused by the pandemic.

Author’s response: Women in our study were examined in October – November and for men as far as possible we maintained the narrow time window of data collection. To be more specific, 96% of all men were examined between August-December, of which 82% during August-October, so the time span was not wide. Nonetheless, as reviewer suggested, we checked whether in our sample of men perceived stress had a decreasing trend at the longer time after outbreak of pandemic. The MEANs (SDs) of PSS scores in men equal:14.67 (3.1), 11.83 (5.8), 15.68 (5.5), 18.78 (6.1), 17.56 (7.5), 16.12 (9.9) consecutively from July to December, without statistically significant difference between months and any evidence of monotonic, or even more so decreasing, trend.

The fact that, when you describe the chapter MATERIALS AND METHODS, you do not talk about this chapter but refer to reference 29, makes it difficult to follow the study. The same happens in the dietary assessment chapter when you refer to references 26 and 31.

Please describe the above-mentioned chapters in more detail, remembering that this manuscript is the result of a larger study that took shape in several articles.

Author’s response: Thank you for such a crucial suggestion. We extended the Materials and methods section and we added directly in the text the most important issues (please see lines 131-165: “Only individuals adhering to (...)”, describing the main information contained in reference 34 (previously 29) and lines 194-209:“Based on 7-day food diaries (...)” describing the main information contained in references 27 and 36 (previously 26 and 31)).

Also, please explain, in more detail, why you used these inclusion criteria and not others.

Author’s response: We wanted to examine how a diet (with a varied content of phytochemicals) influences the risk of occurrence of the endpoints in the quite homogeneous group of people of generally good health, as far as possible eliminating the effect of interference by other confounders. Therefore we recruited only: (i) early adults/middle aged people (25-45 years – please see point 2); (ii) subjects with non-extreme values of bmi – people who were underweight or obese were excluded as potentially having weaker immunity and more exposed to some serious health consequences (for young people with overweight we did not find incontrovertible evidence prompting us to remove them from the research).

In table 1, the average for the duration of sleep($) for all participants is not correctly expressed (0.3 H/d ????)

Author’s response: Thank you for such a detailed revision and the notice of our mistake, we corrected it by transforming the values previously expressed in days to hours.

Many questions are included in the title of figure 2. These could only be written in manuscript without crowding the title of figure 2.

Author’s response: We agree with the Reviewer's suggestion. The contents of individual questions have only been left in the text of the manuscript and in Supplementary Materials, at the same time they have been removed from the chart description.

In the manuscript you referred to the attached supplementary materials, but I could not find them. Only the current manuscript is attached.

Author’s response: We are grateful for the information, very sorry for the inconvenience. During article's submission we included the supplementary file together with the text of the main manuscript. We suppose that some technical error occurred and we hope that in the first round of the revision the problem will be resolved.

I appreciate that the limitations of the article are well presented

Good luck!

Author’s response: We are grateful for your kind words and the positive evaluation of our work.

We did our best to improve the manuscript, we hope that we have met your expectations. Once again, we sincerely thank you for your assistance and feedback.

Reviewer 2 Report

Comments and Suggestions for Authors

The study examines the association between lignan and phytosterol intake and the level of perceived stress in a group of young people during the COVID-19 pandemic in Poland. Based on a mixed diet high in lignans and plant sterols and low in animal products, similar to the Mediterranean dietary pattern, the authors confirmed that specific lignans and phytosterols were correlated with the level of perceived stress. In conclusion, both lignans and phytosterols may help to prevent or reduce stress through their antioxidant, anti-inflammatory, and immunostimulatory effects. They may also play a beneficial role in the management of depressed mood states in young people and may be an excellent alternative to conventional antidepressants, avoiding the adverse side effects of these drugs.

The MS is well organized and interesting. The authors use previously developed and published methods. The paper is well-written and readable. Statistical analyses are adequately described. The strengths and the limitations are discussed.

Please note the following.

1.       It is not clear how long the participants stayed on the diet. Was it just 7 days or longer?

2.       Why is there such a difference in protocol between men and women? (July-December vs. October-November). Were the men's data collected all at once or between July and December?

3.       What was the reason for including overweight people (BMI 25-29.9) in the trial? Was any relationship found between dietary changes and body weight?

4.       What was the main source of lignans and plant sterols in the diet? It would be worth mentioning the main sources and how they differ from the daily diet, not just referring to literature.

5.       The questions in the correlation analysis (section 3.2) refer to the period of 1 month, how does it relate to the period during which the diet was implemented?

6.       Was there a  group with no change in diet in the trial? What were the levels of lignans and phytosterols before the diet was changed to compare the effects of the diet enrichment effect?

7.       Lignans are one of the classes of phytoestrogens postulated to have antidepressant effects. Due to their oestrogenic effects, was a separate analysis performed taking into account gender and the compounds investigated?

In Table 1 - Sleep duration H/d is 0.3 (0.0), please check this.

Line 477 in vitro-  in italic.

Author Response

Dear Editor,

We would like to express our appreciation to the reviewers and editorial board for taking the time and effort dedicated to review our manuscript. The reviewers’ comments were very useful and provided insightful guidelines that helped us to improve the scientific quality of the study. We have carefully considered your suggestions. Here we submit a point-by-point response to the questions. All responses for reviewers are in red and main changes to manuscript are highlighted in the manuscript text.

We paste the answers both, here in the designated window below, as well as in the file attached.

RESPONSE TO REVIEWER 2 COMMENTS

The study examines the association between lignan and phytosterol intake and the level of perceived stress in a group of young people during the COVID-19 pandemic in Poland. Based on a mixed diet high in lignans and plant sterols and low in animal products, similar to the Mediterranean dietary pattern, the authors confirmed that specific lignans and phytosterols were correlated with the level of perceived stress. In conclusion, both lignans and phytosterols may help to prevent or reduce stress through their antioxidant, anti-inflammatory, and immunostimulatory effects. They may also play a beneficial role in the management of depressed mood states in young people and may be an excellent alternative to conventional antidepressants, avoiding the adverse side effects of these drugs.

The MS is well organized and interesting. The authors use previously developed and published methods. The paper is well-written and readable. Statistical analyses are adequately described. The strengths and the limitations are discussed.

Author’s response: We thank the Reviewers for the time dedicated to review our manuscript and for valuable comments.

Please note the following.

  1. It is not clear how long the participants stayed on the diet. Was it just 7 days or longer?

Author’s response: Thank you for the valid remark, it was just 7 days. Taking into account the Reviewer's request we extended the Materials and methods section and we described in more detail this point (please see lines 131-165: “Only individuals adhering to (...)”). In particular, we clarified the issue by adding the sentences: “During 7 consecutive days, participants registered their dietary records and physical activity using the Polar M430 (Polar Electro Oy, Kempele, Finland) watch received on the first visit. The participants received careful education on how to keep the dietary records for one week of observation using e.g. kitchen scales or professional website showing portion sizes of food and the amounts in grams.”

  1. Why is there such a difference in protocol between men and women? (July-December vs. October-November). Were the men's data collected all at once or between July and December?

Author’s response: In  the case of both arms of the study, for men and women, the same design was implemented. However, the study on women started a little later, in October 2020, while examination of 97% men began in August (except for 3 men examined in July). The choice to initiate the study for men and women in different months came from financial and organisational reasons, involving the logistics to carry out a research with the financial resources available at the time. Finally the duration of the study was also affected by pandemic restrictions. All these issues resulted in shortening of the study, especially in the case of females. However, generally the narrow time window of data collection was maintained. Taking into account importance of the Reviewer’s remark we added in the limitations of the study the sentence “Second, due to economic restrictions we had to stop the use of face-to-face interviews during the second phase of the study, which resulted in a predominance of men among participants as well as we did not conduct longitudinal observation of eating habits and stress” (please see lines 618-623 of the discussion).

  1. What was the reason for including overweight people (BMI 25-29.9) in the trial? Was any relationship found between dietary changes and body weight?

Author’s response: Thank you for the question. We wanted to examine how a diet (with a varied content of phytochemicals) influences the risk of occurrence of the endpoints in the quite homogeneous group of people of generally good health, as far as possible eliminating the effect of interference by other confounders. Therefore we recruited only: (i) early adults/middle aged people (25-45 years – please see point 2); (ii) subjects with non-extreme values of bmi – people who were underweight or obese were excluded as potentially having weaker immunity and more exposed to some serious health consequences (for young people with overweight we did not find incontrovertible evidence prompting us to remove them from the research).

  1. What was the main source of lignans and plant sterols in the diet? It would be worth mentioning the main sources and how they differ from the daily diet, not just referring to literature.

Author’s response: As suggested by the Reviewer, we have supplemented the discussion text with the main sources of lignans and plant sterols in the diet. We have also added a section on the differences in diets with the lowest (daily diet, similar to the Western pro-inflammatory diet) and highest phytochemical content (diet similar to the Mediterranean diet) in our study.

  1. The questions in the correlation analysis (section 3.2) refer to the period of 1 month, how does it relate to the period during which the diet was implemented?

Author’s response: Thank you for such a detailed revision and for pointing out the difference between  the time of tracking the diet (7 days) and the time covered by the question regarding perceived stress (the last month). The research plan assumed one week observation of the diet within the time frame of the entire project. It was considered as an adequate period to get to know the eating habits of the participants (in both weekdays and weekends), especially since they were educated not to change their usual diet during the study (please see lines 131-165: “Only individuals adhering to (...)” in the manuscript). At the same time we intended to limit the number of participants’ dropping out of the project (keeping food diaries is a time consuming task and participants could have difficulties with recording diaries with high rigour for a longer time). After a week of dietary observation, in the second visit, subjects reported the level of stress perceived during last month, including the recent week when food diaries were recorded. Participants were requested to take into account the average perceived stress level and eliminate atypical, extremely stressful days. We agree with the Reviewer that this element should be clarified and the additional comment were made in the manuscript as limitation (discussion section, lines 631-636), namely “Moreover, the questionnaire assessed (…)”

  1. Was there a  group with no change in diet in the trial? What were the levels of lignans and phytosterols before the diet was changed to compare the effects of the diet enrichment effect?

Author’s response: Thank you very much for this question. The current study reports on the data from an observational study design, not intervention study, thus we do not have the data for pre post measurements. We enrolled volunteers following their usual diet (vegetarian or traditional). We did not make any dietary modifications or variations during the examination time and we asked the participants to do the same (please see lines 131-165: “Only individuals adhering to (...)” in the manuscript). Therefore, we did not assess changes in lignan and phytosterol levels, assuming that participation in the study did not result in any dietary changes.

  1. Lignans are one of the classes of phytoestrogens postulated to have antidepressant effects. Due to their oestrogenic effects, was a separate analysis performed taking into account gender and the compounds investigated?

Author’s response: Thank you for the remark. The number of women included in the study (26 persons) was too small to perform separate analysis in sex strata. However, we adjusted all analyses to sex by including it as a potential confounder. Similarly, we tested various models controlling the results to sets of covariates such as alcohol drinking, smoking status, age, physical activity, BMI, and body fat. The results were very stable, regardless of level of standardisation, showing robustness of our findings. Taking into account valid Reviewer’s suggestion we added in conclusion “Further research involving larger sample of women is recommended, allowing to conduct sex-strata analysis (...)” (please see lines 657-658).

  1. In Table 1 - Sleep duration H/d is 0.3 (0.0), please check this.

Author’s response: Thank you for such a detailed revision and the notice of our mistake, we corrected it by transforming the values previously expressed in days to hours.

  1. Line 477 in vitro-  in italic.

Author’s response: Corrected.

We have done our best to improve the article and we hope that we have met your expectations. Thank you once again for your feedback and assistance.

Reviewer 3 Report

Comments and Suggestions for Authors

Dear Authors

This manuscript investigated negative association of lignan and phytosterol intake with stress perception during the COVID-19 pandemic for a Polish study on young adults. The manuscript is well-written and the study has a good rationale and sound hypotheses. This study appropriately complements and extends similar studies published worldwide previously.

Namely, it is very interesting study and it presents good data on this very important topic. I believe that this is very excellent issue in field of nutrition and public health section. Moreover, this topic was not clearly revealed in the field of nutrition and public health section. However, it has some minor issues.

Minor issue

Abstract: (1) Sort alphabetically in Keywords

(2) PSS -> Perceived Stress Scale

Introduction

Line 64: WHO -> World Health Organization (WHO)

Abbreviations should be defined in the first instance in whole manuscript.

Please introduce about association between lignans and phytosterol intake and the level of perceived stress focused on other aged group (e.g. elderly) and other countries (e.g. USA, UK).

Methods

Line 123 BMI -> body mass index (BMI)

Please refer to example. SPSS for Windows (version 23.0; IBM Corp., Armonk, NY, USA).

Line 126 Polar M430 watch and Line 155 Dieta 6.0 program -> insert information, Polar M430 watch (???, ???, ???, city, country)

Results

Change from “Kg/m2” to “Kg/m2” in whole manuscript

Line 276 OR, 95% CI -> odd ratio (OR), 95% confidence interval (CI)

T1, T2, T3, too.

Abbreviations should be defined in the first instance in whole manuscript.

Discussion

The discussion section is well-written.

Please add some sentences of application in this field (nutrition and public health).

Comments on the Quality of English Language

I recommend that this manuscript should be edited by an English professional editor for more readable. There are several typo and grammatical errors.

Author Response

Dear Editor,

We would like to express our appreciation to the reviewers and editorial board for taking the time and effort dedicated to review our manuscript. The reviewers’ comments were very useful and provided insightful guidelines that helped us to improve the scientific quality of the study. We have carefully considered your suggestions. Here we submit a point-by-point response to the questions. All responses for reviewers are in red and main changes to manuscript are highlighted in the manuscript text.

We paste the answers both, here in the designated window below, as well as in the file attached.

RESPONSE TO REVIEWER 3 COMMENTS

Dear Authors

This manuscript investigated negative association of lignan and phytosterol intake with stress perception during the COVID-19 pandemic for a Polish study on young adults. The manuscript is well-written and the study has a good rationale and sound hypotheses. This study appropriately complements and extends similar studies published worldwide previously.

Namely, it is very interesting study and it presents good data on this very important topic. I believe that this is very excellent issue in field of nutrition and public health section. Moreover, this topic was not clearly revealed in the field of nutrition and public health section. However, it has some minor issues.

Author’s response: We thank the Reviewers for the time dedicated to review our manuscript and for valuable comments.  We are grateful for kind words and the positive evaluation of our work.

Minor issue

Abstract: (1) Sort alphabetically in Keywords

Author’s response: We listed the keywords in alphabetic order.

(2) PSS -> Perceived Stress Scale

Author’s response: We provided the explanation for abbreviation in the abstract (please see line 32).

Introduction

Line 64: WHO -> World Health Organization (WHO)

Author’s response: We provided the explanation for abbreviation as suggested (please see lines 65-66).

Abbreviations should be defined in the first instance in whole manuscript.

Author’s response: Thank you for the observation. As suggested, we revised the abbreviations along the manuscript.

Please introduce about association between lignans and phytosterol intake and the level of perceived stress focused on other aged group (e.g. elderly) and other countries (e.g. USA, UK).

Author’s response: A review of the available knowledge in the literature shows that there is a lack of studies on the effects of lignans and phytosterols on perceived stress levels. However, following the right suggestion of the reviewer, we added a short piece of information to the Introduction section based on several intervention trials and studies performed on animal models (lines 105-115, “Several studies on animal models (...)”). Additionally, we used a reference to the Mediterranean diet characterised by a high content of bioactive compounds, including lignans and phytosterols, as a promising strategy for the prevention and treatment of depression. We introduced notes on its impact on depression in children, adults and the elderly from various countries including the US and the UK (please see lines 600-609).

Methods

Line 123 BMI -> body mass index (BMI)

Author’s response: We explained the abbreviation as suggested (lines 134-135).

Please refer to example. SPSS for Windows (version 23.0; IBM Corp., Armonk, NY, USA).

Line 126 Polar M430 watch and Line 155 Dieta 6.0 program -> insert information, Polar M430 watch (???, ???, ???, city, country)

Author’s response: Thank you for the valid observation. As suggested, we revised the form of writing product’s names along the manuscript., e.g. Polar M430 (Polar Electro Oy, Kempele, Finland) (line 143-144), a calibrated segment analyzer (Tanita BC-418 MA, Tokyo, Japan) (line 156), a portable stadiometer (SECA 213, Hamburg, Germany) (line 162).

Results

Change from “Kg/m2” to “Kg/m2” in whole manuscript

Author’s response: As suggested, we changed  “Kg/m2” to “Kg/m2”.

Line 276 OR, 95% CI -> odd ratio (OR), 95% confidence interval (CI)

Author’s response: The abbreviation for OR and CI was first introduced in the methods section (please see lines 237-238 in the current version of the manuscript).

T1, T2, T3, too.

Abbreviations should be defined in the first instance in whole manuscript.

Author’s response: W introduced the abbreviation for tertiles at first use in the methods section (line 235). Also, as suggested, we revised the abbreviations along the manuscript.

Discussion

The discussion section is well-written.

Author’s response: We are very grateful for the positive evaluation of our work.

Please add some sentences of application in this field (nutrition and public health).

Author’s response: Thank you for the reviewer's fair comment. We have added a section on the use of a diet rich in phytosterols and lignans as an excellent strategy to counteract inflammation associated with the onset and severity of psychiatric disorders and may improve public health  (please see lines 610-619 “It therefore appears that altering dietary patterns (...)”).

Comments on the Quality of English Language

I recommend that this manuscript should be edited by an English professional editor for more readable. There are several typo and grammatical errors.

Author’s response: As suggested, we revised the entire manuscript in context of typos and grammatical errors.

We did our best to improve the manuscript, we hope that we have met your expectations. Once again, we sincerely thank you for your assistance and feedback.
